# Prevention of Delayed Graft Function in Kidney Transplant Recipients through a Continuous Infusion of the Prostaglandin Analogue Iloprost: A Single-Center Prospective Study

**DOI:** 10.3390/biomedicines12020290

**Published:** 2024-01-26

**Authors:** Massimiliano Veroux, Floriana Sanfilippo, Giuseppe Roscitano, Martina Giambra, Alessia Giaquinta, Giordana Riccioli, Domenico Zerbo, Daniela Corona, Massimiliano Sorbello, Pierfrancesco Veroux

**Affiliations:** 1General Surgery Unit, Azienda Policlinico San Marco, University of Catania, 95124 Catania, Italy; floriana.sanfilippo@gmail.com (F.S.); giordanariccioli@me.com (G.R.); 2Vascular Surgery and Organ Transplant Unit, Azienda Policlinico San Marco, University of Catania, 95124 Catania, Italy; giuseppe.roscitano@virgilio.it (G.R.); giambramartina@gmail.com (M.G.); alessia.giaquinta@gmail.com (A.G.); domenico.zerbo@libero.it (D.Z.); coronadany@libero.it (D.C.); pveroux@unict.it (P.V.); 3Intensive Care Unit, Ragusa Hospital, 97100 Ragusa, Italy; maxsorbello@gmail.com

**Keywords:** kidney transplantation, marginal donor, cold ischemia time, donor age, acute kidney injury, deceased donor, graft survival, patient survival

## Abstract

Background: Delayed graft function (DGF) is common after kidney transplantation from deceased donors and may significantly affect post-transplant outcomes. This study aimed to evaluate whether an innovative approach, based on the administration of the intravenous prostaglandin analogue iloprost, could be beneficial in reducing the incidence of DGF occurring after kidney transplantation from deceased donors. Methods: This prospective, randomized (1:1), placebo-controlled study enrolled all consecutive patients who received a kidney transplant from a deceased donor from January 2000 to December 2012 and who were treated in the peri-transplant period with the prostaglandin analogue iloprost at 0.27 μg/min through an elastomeric pump (treatment group) or with a placebo (control group). Results: A total of 476 patients were included: DGF was reported in 172 (36.1%) patients in the entire cohort. The multivariate analysis showed that the donor’s age > 70 years (OR 2.50, 95% confidence interval (CI): 1.40–3.05, *p* < 0.001), cold ischemia time > 24 h (OR 2.60, 95% CI: 1.50–4.51, *p* < 0.001), the donor’s acute kidney injury (OR 2.71, 95% CI: 1.61–4.52, *p* = 0.021) and, above all, the recipient’s arterial hypotension (OR 5.06, 95% CI: 2.52–10.1, *p* < 0.0001) were the strongest risk factors for developing post-transplant DGF. The incidence of DGF was 21.4% in the treatment group and 50.9% in the control group (*p <* 0.001). Interestingly, among patients who developed DGF, those who received iloprost had a shorter duration of post-transplant DGF (10.5 ± 8.3 vs. 13.4 ± 6.7, days, *p =* 0.016). Conclusions: This study showed that the use of a continuous infusion of iloprost could safely and effectively reduce the incidence of DGF in recipients of deceased-donor kidneys, allowing a better graft functionality as well as a better graft survival.

## 1. Introduction

Kidney transplantation is the best available therapy for end-stage renal disease. Recent improvements in short-term survival after kidney transplantation have been observed as a consequence of more effective immunosuppressive agents and improved perioperative management. However, this short-term success has not led to an equal improvement in long-term outcomes [1]. The great disparity between the organs available for kidney transplantation and the number of patients on waiting lists, as well as the progressive increase in donor age, has led most transplant centers to expand their acceptance criteria by more and more frequently including the use of expanded-criteria donors, including donors with acute kidney injury (AKI) [2,3,4,5,6]. The absence of immediate function, known as delayed graft function (DGF), commonly defined as the need for dialysis during the first week after transplantation, is a well-known risk factor for worse graft function in kidney transplantation [7,8,9,10,11], even in the absence of acute rejection [8,11,12]. DGF has an incidence ranging between 10% and 50% of deceased-donor kidney transplantations [7,8,10,11,12,13,14,15,16,17], but its incidence may raise to 63% in kidney transplants from marginal donors [18]. The most important risk factors for DGF are older donor age, donor acute kidney injury, older recipient age and prolonged cold ischemia time [7,8,10,11,12,13,14,15,16,17]. The donor’s age has a clear impact on the incidence of DGF and, therefore, to reduce the incidence of DGF in kidney transplantation from marginal donors, many centers have adopted a policy of local allocation of kidneys to pre-consented candidates, in most cases obviating biopsy, and by reducing the cold ischemia time, they lowered the incidence of DGF [19,20]. An animal model of renal ischemia and reperfusion suggests that DGF may have an immunological basis due to an increase in endothelial lymphocyte interaction and resulting vasospasm [9]. So far, different therapeutic modalities have been proposed, ranging from the use of calcium-channel blockers to prostaglandin E_1_ infusions or non-selective endothelin receptor antagonists, all with debatable results [15,21]. Iloprost is a prostaglandin analogue that has been used in the treatment of peripheral arterial disease thanks to its vasodilatory and antiaggregant effects. Iloprost exerts its vasodilatory effect through prostaglandin pathways and may be beneficial in preventing ischemia/reperfusion injury thanks to the inhibition of leukocyte activation and adhesion, free oxygen radicals and proteolytic enzyme secretion, finally resulting in endothelial protection [22]. Because of the lack of effective therapies and given the impact on graft survival of delayed graft function, this study evaluated if an innovative approach based on the administration of the intravenous prostaglandin analogue iloprost could be beneficial in reducing the incidence of DGF after kidney transplantation from deceased donors.

## 2. Materials and Methods

### 2.1. Study Population

In this prospective, randomized (1:1), placebo-controlled study, we retrospectively evaluated the incidence of DGF and the long-term outcomes in a cohort of consecutive patients who received a kidney transplantation from a brain-death-deceased donor and who were treated in the peri-transplant period with the prostaglandin analogue iloprost (Italfarmaco, Milan, Italy) (treatment group) or a placebo (control group). Eligible patients were randomly assigned to one of two experimental groups (treatment group or control group) in a 1:1 ratio through a random allocation sequence. Delayed graft function was defined as the need for at least one dialysis session within one week after transplantation [10,11]. Primary kidney nonfunction was defined as the complete lack of functionality, in that the recipient never discontinued their dialysis sessions after transplantation.

This study was approved by the Local Ethic Committee of the Azienda Policlinico San Marco of the University of Catania, and the patients provided written informed consent to undergo renal transplantation and participate in this protocol. Marginal donors were identified based on one or more of the following characteristics: age > 60 years; history of long-standing (>10 years) diabetes and/or hypertension; and terminal serum creatinine > 2 mg/dL [23]. All kidneys > 70 years were histologically evaluated through a tissue sample obtained from the superior pole of each donor’s kidney. Kidneys from which a biopsy specimen was obtained were selected and allocated based on the severity of chronic changes, which was quantified based on a predefined histologic score [3]. In brief, all changes in each evaluated component of the kidney tissue (vessels, glomeruli, tubules and connective tissue) received a score ranging from 0 to 3. Each received a score of 0 if no changes were observed, and a score of up to three was given if marked changes were present. Kidneys with a global score ranging from 0 to 3 were considered for use as a single transplant, and those with a score from 4 to 6 were considered for use as a dual transplant; those with a score of 7 or greater were discarded [3]. Kidney transplantations were performed by the same surgical team with a standardized surgical procedure, as previously described [24]. A total of 476 consecutive patients, who underwent a kidney transplant from deceased donors between January 2000 and December 2012, met the inclusion criteria and were prospectively randomized on a 1:1 basis at the time of transplantation. A central venous line was placed in all patients at the time of surgery and was removed on the seventh postoperative day. The anesthesiologic protocol was standardized for all patients: anesthesia was induced with propofol (2 mg · kg^−1^), fentanyl (1.5 mcg · kg^−1^) and cisatracurium (2 mg · kg^−1^); after tracheal intubation, anesthesia was maintained with cisatracurium and fentanyl on demand and sevoflurane (1–1.5 MAC). All patients received the same amount and type of infused fluids (emagel (500 mL) followed by saline (1000 mL) and 5% glucose up to monorenal phase), calculated as 10 mL · kg^−1^ · h^−1^, starting immediately after the central venous line placement [25]. Patients in the treatment group intravenously received iloprost at 0.27 microg/min through an elastomeric pump, as previously described [22], beginning from the oro-tracheal intubation of the patients, while patients in the control group received a saline solution. Patients with a known intolerance or a previous side effect related to the treatment with iloprost were excluded from the study (2 patients). The infusion of iloprost was continued for 72 h in patients without DGF, while in patients with DGF, it was maintained until the restoration of renal function, defined as the discontinuation of dialysis treatment. Acute kidney injury in the kidney donor was defined as a ≥50% increase in the last serum creatinine level from the level of the day of admission [26,27]. Recipient hypotension was defined as the presence of a systolic arterial pressure < 100 mmHg at the time of admission for the kidney transplantation. Refractory hypotension was treated with extra colloids and a dobutamine infusion titrated to effect, as previously reported [25]. Patients in both the treatment group and the control group received 100 mg of furosemide at the time of reperfusion. The immunosuppression protocol was based on a triple-drug maintenance therapy with/without an induction therapy with basiliximab or thymoglobuline, as previously reported [24]. In all patients in whom an acute rejection was suspected on the basis of the worsening of graft function and a rise in serum creatinine levels, a graft biopsy was obtained, and the rejection was scored according to the Banff classification [28]. Rejection therapy consisted in steroid pulses of 500 mg of methylprednisolone for three days.

### 2.2. Statistical Analysis

This study included all recipients of deceased-donor kidneys recruited over a 12-year period. The characteristics at the time of transplant of the entire cohort were compared with the use of Fisher’s exact test, the chi-square test and Student’s *t* test. The primary analysis was a comparison of the incidence of delayed graft function between recipients treated with an intravenous infusion of iloprost and a group of recipients as the control group. The primary end-point (delayed graft function) and secondary end-points were evaluated with the use of a Cox regression model that included the donor’s age, the donor’s terminal serum creatinine, the donor’s hemodynamic parameters (stay in ICU, presence of hypotension, presence of acute kidney injury (AKI)), donor–recipient arterial pressure match, the recipient age and the rate of acute rejection. Data are expressed as mean ± standard deviation (SD). To compare parametric variables, the Pearson chi-square test or Fisher’s exact test was used. To compare nonparametric variables, Student’s *t*-test or the Mann–Whitney U test was used. The difference between the two means was calculated using the Wilson test. Odds ratios (ORs) were reported with a 95% confidence interval (95% CI) and *p*-values. The level of statistical significance was determined at *p* < 0.05. Predictive factors of DGF with a *p*-value < 0.5 in the univariate analysis were considered for the multivariate model using a downward stepwise binary logistic regression analysis. The rates of graft survival, censored for death, were plotted with the Kaplan–Meier method. Statistical analyses were performed with the SAS software version 9.2 and Microsoft Excel 2021.

## 3. Results

A total of 476 kidney transplant recipients were collected in the study period.

The mean age was 48. 5 ± 18.5 years with a prevalence of male recipients (301 patients, 63.2%), with a mean BMI of 26.6 ± 9.5 kg/m^2^. The most frequent cause of end-stage renal disease was autosomal dominant polycystic kidney disease (97 patients, 20.3%), followed by glomerulonephritis (49 patients, 10.2%) and diabetes (24 patients, 5%), while the cause of ESRD was unknown in 169 (35.5%) of patients. Hemodialysis was the most common renal replacement therapy (454 patients, 95.3%), while eight patients (1.6%) received a pre-emptive kidney transplantation. The mean time on dialysis was 49 ± 24.3 months, while the mean time on the waiting list was 19.4 ± 12.8 months. In the entire cohort, the mean donor age was 50.8 ± 26.4 years, and 186 recipients (39%) received a kidney graft from donors aged > 55 years. A total of 25 donors (5.2%) had diabetes and 150 (31.5%) were hypertensive. Seventy-nine donors (16.5%) had a terminal serum creatinine > 1.5 mg/dL.

Delayed graft function was reported in 172 (36.1%) patients (Table 1).

The comparative analysis showed that kidney transplant recipients who received a kidney graft from a donor aged > 70 years, with cold ischemia > 24 h and with an acute kidney injury were at higher risk of developing delayed graft function (Table 1). A longer time on the waiting list and a longer time on dialysis were also significant risk factors for DGF. Moreover, recipients aged > 60 years with arterial hypotension were more subjected to develop post-transplant DGF. Both the donor and recipients’ gender and the donor’s use of vasoactive amines, as well as the stay in the ICU, did not have an impact on the incidence of DGF. The multivariate analysis (Table 2) confirmed that donor age > 70 years (OR 2.06, 95% CI: 1.40–3.05, *p* < 0.001), cold ischemia time > 24 h (OR 2.60, 95% CI: 1.50–4.51, *p* < 0.001), donor AKI (OR 2.71, 95% CI: 1.61–4.52, *p* = 0.021) and, above all, recipient arterial hypotension (OR 5.06, 95% CI: 2.52–10.1, *p* < 0.0001) were the strongest risk factors for developing post-transplant DGF.

The Kaplan–Meier analysis demonstrated a significantly better survival among patients with immediate graft function compared with patients who developed DGF (*p* < 0.05) (Figure 1).

A subsequent analysis was performed to evaluate if the intravenous treatment with iloprost could reduce the incidence of DGF compared to a placebo: 238 patients in the treatment group and 238 in the control group were included in the analysis (Table 3).

There was no significant difference in terms of donor characteristics (age, gender, terminal serum creatinine, incidence of diabetes and arterial hypertension, stay in ICU and cause of brain death) between the two groups, but the treatment group had a longer cold ischemia time. The recipient characteristics (age, gender, time on dialysis, rate of primary non-function and acute rejection) and type of immunosuppression were similar between the two groups, but recipients in the treatment group had a longer time on the waiting list (23.9 ± 21 vs. 15 ± 12 months, *p* < 0.001) compared with the control group.

The incidence of DGF was 21.4% in the treatment group and 50.9% in the control group (*p* < 0.001), suggesting that the use of iloprost could significantly reduce the incidence of such a complication. The use of iloprost was well tolerated and only four patients (1.6%) required a treatment discontinuation: two patients reported flushing and headaches, while in two patients, the treatment was stopped due to severe arterial hypotension. All patients were also evaluated for liver function during the entire follow-up and no sign of iloprost-related liver toxicity was detected.

Interestingly, among patients who developed DGF, patients receiving iloprost had a shorter duration of post-transplant DGF (10.5 ± 8.3 vs. 13.4 ± 6.7, days, *p* = 0.016)

Patients in the treatment group had a significantly lower hospital stay (10.5 ± 4.4 vs. 13.3 ± 6.4, *p* < 0.05) and a better 1 year (1.41 ± 0.61 vs. 1.60 ± 0.65 mg/dL, *p* = 0.008) and 5 years (1.50 ± 0.62 vs. 1.66 ± 0.81 mg/dL, *p* = 0.045) serum creatine, but the 10 y serum creatinine was similar between the two groups. The 1-year (93.3% vs. 88%, *p* < 0.05) and 5-year (83% vs. 79%, *p* < 0.05) graft survival was significantly better in the treatment group compared with the controls, respectively. The 1-year, 5-years and 10-years patient survival was similar between the two groups.

## 4. Discussion

This prospective study showed that the use of the prostaglandin analogue iloprost was safe and effective in reducing the incidence of DGF in kidney transplant recipients by acting on its development mechanisms. Delayed graft function is still a common complication after kidney transplantation, and the increasing donor and recipient ages observed in the last decade may render transplant programs reluctant to utilize kidneys at a higher risk of developing DGF. The full mechanism of DGF is still not clear in detail. The tubular damage resulting from ischemia/reperfusion injury seems the basis of the decreased glomerular impairment of DGF. Tubular epithelial cell degeneration, tubular cell exfoliation, interstitial edema and interstitial cellular infiltration may cause, in the early phase, a tubular obstruction, resulting in a low net filtration pressure. Later, decreased sodium reabsorption results in afferent vasoconstriction and diminished glomerular filtration pressure through the tubuloglomerular feedback mechanism [6,10,15]. A recent meta-analysis showed [10] that DGF is associated with significantly worse short- and long-term outcomes post-transplant, including increased graft failure, acute allograft rejection and, in single-center studies, 1-year mortality. Moreover, kidney transplant recipients who experienced DGF had an overall reduction of 5.46 mL/min in eGFR at 1-year post-transplant compared with those who did not experience DGF. Several studies suggested that the presence of DGF implies a poor outcome, and this could be amplified when receiving a kidney from a marginal donor, suggesting that the effect of donor-dependent damage may be amplified by ischemia reperfusion injury [9]. This was confirmed in this study: the overall incidence of DGF in this cohort was 36.1%, and the incidence of DGF was higher in patients receiving an older graft with a long ischemia time and with kidney injury, suggesting that these grafts may be more vulnerable to ischemia/reperfusion injury, which may further reduce the nephron mass. Furthermore, donors aged > 70 years, cold ischemia time > 24 h and donor AKI were significantly associated with an increased risk of developing DGF. Interestingly, the recipient’s hypotension, especially when receiving a kidney from a hypertensive donor, was an independent risk factor for DGF. Indeed, hypertensive marginal donors present frequently atherosclerotic stenosis of the renal arteries, meaning that a low mean arterial pressure in the recipient would not be able to ensure a stable and viable perfusion of the graft, amplifying the ischemic damage of the ischemia/reperfusion injury. Therefore, a special care must be taken in the allocation of marginal kidneys from hypertensive donors by avoiding, if feasible, recipients with a low mean arterial pressure. DGF may, alternatively, through an increased expression of MHC antigens, determine an inflammatory response with the release of pro-inflammatory cytokines in tubular cells, which may finally contribute to an increased rate of rejection in ischemically injured kidneys [15], and this may reflect a worse outcome [10,29,30]. In this regard, it has been shown in an experimental model [31] that the development of ischemia/reperfusion injury is closely related to cold preservation, which may increase vasoconstriction, and that the addiction of anti-ischemic drugs may prevent renal injury and had a significant beneficial effect on renal function. The prostaglandin I_2_ stable analogue iloprost has been approved for the therapy of lower-limb critical ischemia secondary to peripheral obliterative disease [22], and it exhibits a protective effect against renal ischemia/reperfusion injury in rabbit models [32,33]. Ischemia/reperfusion injury results in the release of free radicals and induces an increase in renal tissue levels of tumor necrosis factor alpha and cytokine-induced neutrophil chemoattractant. This may lead to an activated neutrophil-induced endothelial cell injury, which, finally, might lead to the ischemia of the proximal tubules [6,15]. Iloprost has demonstrated cytoprotective properties by inhibiting intracellular lysosome distribution, which is implicated in membrane destabilization and the release of free radicals [34,35]. Moreover, iloprost, by inhibiting platelet aggregation, oxygen free radical production and neutrophil activation, could improve ischemia reperfusion-induced injury by improving renal microcirculation [32,33]. In clinical settings, iloprost failed to improve cyclosporin-induced renal hypoperfusion in stable renal transplant recipients [36], but in a prospective randomized trial in association with diltiazem, iloprost determined a reduction in delayed graft function in deceased donor kidney transplants [34]. This prospective study confers some new insights: first, it was conducted on a selected population of deceased donor kidney recipients; second, this is the first study in which iloprost was administered via a continuous IV infusion through an elastomeric pump so that the drug may have consistent and prolonged action throughout the entire 24 h period.

The rationale of this study was that by inducing vasodilatation of the micro- and macrocirculation of the renal vasculature, we might reduce the ischemic damage resulting from ischemia/reperfusion injury. The results of this study seem to confirm such an assumption, because the incidence of DGF was significantly reduced in kidney transplant recipients who received iloprost compared with controls (21.4% vs. 50.9%, respectively, *p* < 0.001). This aspect is particularly useful when transplanting kidneys from marginal donors, in whom DGF is associated with a more complicated postoperative management and, finally, with a worse outcome [9,10,11]. The treatment with iloprost resulted in a significantly shorter hospital stay (10.5 ± 4.4 vs. 13.3 ± 6.4, *p* < 0.05) and a significantly better 1-year (1.41 ± 0.61 vs. 1.60 ± 0.65 mg/dL, *p* = 0.008) and 5-years (1.50 ± 0.62 vs. 1.66 ± 0.81 mg/dL, *p* = 0.045) graft function compared with patients who received a placebo. Interestingly, among patients who developed DGF after transplantation, patients treated with iloprost had a significantly shorter duration of post-transplant DGF, suggesting that iloprost could have a beneficial effect on the repair mechanism after ischemia/reperfusion injury, since graft survival and the rate of acute rejection may be linked to the duration of DGF [8]. Patients treated with iloprost had a significantly better 1-year and 5-years graft survival compared with the control group, confirming that the prevention of DGF may have a beneficial effect on graft survival [10,11,37]. However, iloprost had no clinical impact on long-term outcomes: as it is well known, long-term graft survival is strongly related to many factors, including the patient’s comorbidities, the chronic rejection and the development of de novo or recurrent glomerulonephritis, which could not be influenced by the treatment with iloprost. This study has some limitation: the data were analyzed retrospectively, but the study was performed in a prospective fashion, so this is one of the largest studies evaluating the long-term impact of DGF on post-transplant outcomes; second, the two groups were not completely matched and this may cause differences in data analysis; lastly, the treatment with iloprost did not result into any difference in the acute rejection rate. In our center, we do not perform protocol kidney biopsies in kidney transplant recipients, and this practice could have missed some sub-clinical acute rejection episodes in patients with DGF.

In conclusion, this study suggests that iloprost could safely reduce the incidence of delayed graft function in recipients of deceased-donor kidneys. This would result in better graft functionality and a better graft survival. To further reduce the incidence of delayed graft function, some risk factors, such as prolonged cold ischemia time and low recipient mean arterial pressure, should be minimized through a better allocation of deceased-donor kidneys.

## Figures and Tables

**Figure 1 biomedicines-12-00290-f001:**
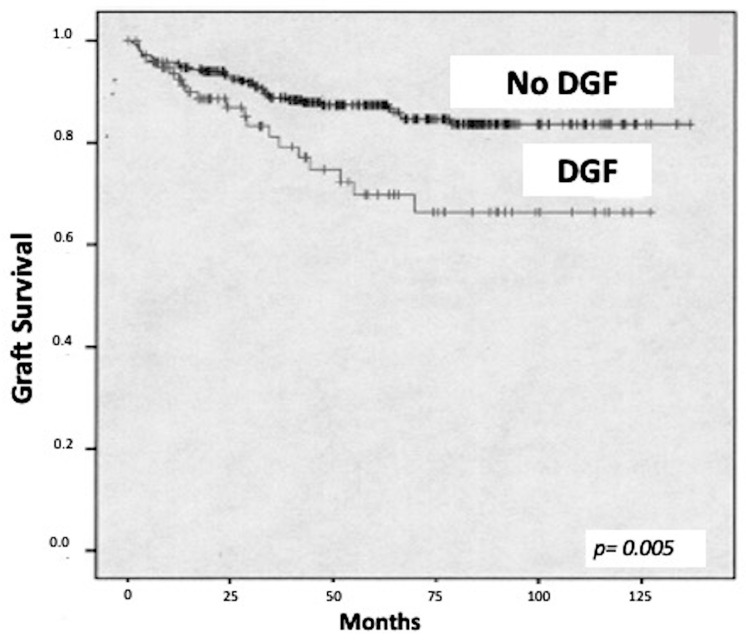
Kaplan–Meier analysis demonstrated a worse graft survival in patients with delayed graft function compared to patients with immediate graft function.

**Table 1 biomedicines-12-00290-t001:** Analysis for the risk factors for delayed graft function in the entire cohort.

	DGF (*n* = 172)	No DGF (*n* = 304)	*p* Value
Mean Donor’s age (year)	57.2 ± 15.8	47.1 ± 18.8	<0.001
Donor’s age			
<55 years	76 (44.1)	191 (62.8)	0.032
66–70 years	49 (28.4)	77 (25.3)	0.543
>70 years	47 (27.3)	36 (11.8)	<0.001
Donors with AKI	41 (23.8)	38 (12.5)	0.001
Female Donor	75 (43.6)	155 (50.9)	0.121
Use of vasopressors	165(95.9)	285 (93.7)	0.744
Stay in ICU	5.4 ± 4.0	4.9 ± 3.9	0.251
Mean Cold Ischemia time (min)	1070 ± 450	882 ± 331	<0.001
Cold ischemia time			
<24 h	132 (76.7)	270 (88.9)	0.635
>24 h	40 (23.3)	34 (11.1)	<0.001
i	52.9 ± 11.4	46.0 ± 11.9	<0.001
Male recipients (%)	110 (63.9)	193 (63.4)	0.884
Recipient’s Hypotension	30 (17.4)	12 (3.9)	<0.001
Waiting list (months)	24.4 ± 21.2	16.5 ± 14.4	0.001
Time on dialysis (months)	65.0 ± 50.8	39.9 ± 32.7	<0.001

**Table 2 biomedicines-12-00290-t002:** Multivariate analysis for the incidence of delayed graft function.

Characteristics	OR	95% CI	*p* Value
Donor age			
Donor age < 70 years	reference		
Donors age > 70 years	2.50	1.40–3.05	<0.001
Cold ischemia time			
<24 h	reference		
>24 h	2.60	1.50–4.51	<0.001
Donors with AKI	2.71	1.61–4.52	0.021
Time on dialysis			
<12 months	reference		
12–24 months	1.01	0.66–1.56	0.908
>24 months	2.87	1.91–4.33	<0.001
Recipient age > 60 years	3.39	2.14–5.38	<0.001
Recipient’s hypotension	5.06	2.52–10.1	<0.001

**Table 3 biomedicines-12-00290-t003:** Clinical characteristics and comparison between the recipients who received iloprost (*n* = 238) and the control group (*n* = 238).

Group and Characteristics	Treatment Group (*n* = 238)	Control Group (*n* = 238)	*p* Value
**Donor**			
Age (year)	50.9 ± 20.4	50.7± 19.8	0.845
Male Sex (%)	91 (38.2)	69 (28.9)	0.032
Terminal Serum Creatinine (mg/dL)	1.13 ± 0.3	1.11 ± 0.3	0.532
Use of vasoactive amines (%)	200 (84%)	203 (85.2)	0.624
Diabetes (%)	18 (7.5)	13 (5.4)	0.223
Arterial Hypertension > 10 years (%)	95 (39.9)	92 (38.6)	0.498
Cold Ischemia Time (h)	17.3 ± 7.4	13.8 ± 6.2	<0.001
Cerebral hemorrhage/ischemia brain death (%)	141 (59.2)	158 (66.3)	0.147
Non-traumatic brain death (%)	93 (39)	78 (32.7)	0.122
Other cause of brain death	4 (1.6)	2 (0.8)	0.554
Use of vasoamine drugs	225 (94.5)	228 (95.7)	0.922
Stay in ICU	4.9 ± 3.8	5.3 ± 4.2	0.279
**Recipient**			
Age (year)	49 ± 11.1	47.9 ± 12	0.324
Male sex (%)	159 (66.8)	142 (59.6)	0.424
Pre-transplant Panel-Reactive Antibody (%)	25 ± 10.2	21 ± 9.7	0.723
Time on Dialysis (mo)	50 ± 23.4	47.1 ± 26.2	0.113
Time on waiting list (mo)	23.9 ± 33	15 ± 16	<0.001
Peritoneal dialysis (%)	5 (2.1)	9 (3.7)	0.433
Dual transplant (%)	17 (7.1)	11 (4.6)	0.115
HCV seropositivity	36 (15.1)	11(4.6)	<0.05
Delayed graft function (%)	51 (21.4)	121 (50.9)	<0.001
Discontinuation of dialysis (day)	10.5 ± 8.3	13.4 ± 6.7	0.016
Primary Non-Function (%)	6 (2.5%)	6(2.5%)	1
Acute rejection	16 (6.7)	25 (10.5)	0.141
Immunosuppression			
Induction (basiliximab)	68 (28.5)	55 (23.1)	0.753
Induction (thymoglobuline)	22 (9.2)	24 (10)	0.883
Tacrolimus	150 (63)	164 (68.9)	0.214
MMF	205 (86.1)	185 (77.7)	0.301
Sirolimus	36 (15.1)	26 (10.9)	0.112
Cyclosporine	29 (12.1)	55 (23.1)	0.108
Everolimus	14 (5.8)	19 (7.9)	0.323
Hospital stay	10.5 ± 4.4	13.3 ± 6.4	<0.05
30-day acute rejection	22 (9.2)	25 (10.5)	0.212
Postoperative Death (30-day)	3 (1.2)	4 (0.8)	0.823
1 year Serum Creatinine (mg/dL)	1.41 ± 0.61	1.60 ± 0.65	0.008
5 years Serum Creatinine (mg/dL)	1.50 ± 0.62	1.66 ± 0.81	0.045
10 years Serum Creatinine (mg/dL)	1.54 ± 0.76	1.64 ± 0.55	0.525
1-year eGlomerular Filtration Rate (mL/min per 1.73 m^2^)	67 ± 15.8	59 ± 12.4	0.012
5-year eGlomerular Filtration Rate (mL/min per 1.73 m^2^)	63 ± 13.4	56 ± 11.8	0.048
10-year eGlomerular Filtration Rate (mL/min per 1.73 m^2^)	61 ± 18.4	55 ± 12.4	0.633
1 year Graft Survival	93.3%	92%	<0.05
5 years Graft Survival	83%	83.7%	<0.05
10 years Graft Survival	75%	74%	0.183
1 year Patient Survival	96.7%	92.1%	0.185
5 years Patient Survival	96.3%	93.7%	0.211
10 years patient Survival	80%	77%	0.172

## Data Availability

De-identified data can be made available upon reasonable request.

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
