# Peer review of "Prevention of Delayed Graft Function in Kidney Transplant Recipients through a Continuous Infusion of the Prostaglandin Analogue Iloprost: A Single-Center Prospective Study"

_biomedicines, 2024, doi:10.3390/biomedicines12020290_

Round 1

Reviewer 1 Report

Comments and Suggestions for Authors

DGF and subsequent poor graft function has been Achilles heel of deceased donor kidney transplantation. Any effort in direction of decreasing the DGF is welcome as it has the potential to improve subsequent graft function. In the present study authors have looked into the role of iloprost which is a prostaglandin analogue in reducing DGF. I have following comments to make regarding the study:

-       What was the proportion of donation after cardiac death and donation after brain death donors in the present study?

-       What were exclusion criteria for giving the drug?

-       The drug did not have any affect on the long term (10 year) graft survival, patient survival. Neither it lead to any different acute rejection rates in short term. Please comment on these.

-       Improvement in DGF did not translate into any difference in acute rejection rates. Please comment on possible reason for same.

-       Please give details of sensitization status of each group

-       Give details of mean HLA mismatches

-       “Data was analyzed retrospectively but the study was conducted in prospective fashion” is a confusing statement

-       Line no.317 – mentions that acute rejection was based on the basis of graft sonography appearance of worsening of graft function. Please remove this line. There are no conclusive ultrasonographic findings to diagnose acute rejection. It is diagnosed by increasing serum creatinine

-       What was the mean tacrolimus level between the two groups?

Comments on the Quality of English Language

Acceptable. Needs minor editing.

Author Response

Dear Reviewer,

I would like to thank you very much for your valuable comments that will improve significantly the scientific value of the manuscript. Please, you will find below the replies to your comments. I hope to have met all your criticisms satisfactorily.

Reviewer 1

DGF and subsequent poor graft function has been Achilles heel of deceased donor kidney transplantation. Any effort in direction of decreasing the DGF is welcome as it has the potential to improve subsequent graft function. In the present study authors have looked into the role of iloprost which is a prostaglandin analogue in reducing DGF. I have following comments to make regarding the study:

-       What was the proportion of donation after cardiac death and donation after brain death donors in the present study?

Thank you for your comment. All kidney transplants were performed from brain death donors. This was specified in the text (line 259)

-       What were exclusion criteria for giving the drug?

The exclusion criteria were a known intolerance, or a side effect related to a previous treatment with iloprost. This was specified in the manuscript (lines 294-5)

-       The drug did not have any affect on the long term (10 year) graft survival, patient survival. Neither it lead to any different acute rejection rates in short term. Please comment on these.

As known, the long term outcomes of kidney transplantation are related to many factors, including the patient comorbities, the chronic rejection and the de novo glomerulonephritis, which could not be influenced by the administration of iloprost

-       Improvement in DGF did not translate into any difference in acute rejection rates. Please comment on possible reason for same.

In our centre, we don’t perform protocol kidney biopsy in kidney transplant recipients, and this practice could have missed some sub-clinical acute rejection episodes in patient with DGF. This was specified in the limits of the study

-       Please give details of sensitization status of each group

This data was inserted in the table

-       “Data was analyzed retrospectively but the study was conducted in prospective fashion” is a confusing statement

Thank you for your comment. The sentence was removed.

-       Line no.317 – mentions that acute rejection was based on the basis of graft sonography appearance of worsening of graft function. Please remove this line. There are no conclusive ultrasonographic findings to diagnose acute rejection. It is diagnosed by increasing serum creatinine

Thank you for your comment. I completely agree with the reviewer and the sentence was modified according to your suggestion.

-       What was the mean tacrolimus level between the two groups?

The mean level of tacrolimus was not reported, since the follow-up time is too long.

DGF and subsequent poor graft function has been Achilles heel of deceased donor kidney transplantation. Any effort in direction of decreasing the DGF is welcome as it has the potential to improve subsequent graft function. In the present study authors have looked into the role of iloprost which is a prostaglandin analogue in reducing DGF. I have following comments to make regarding the study:

-       What was the proportion of donation after cardiac death and donation after brain death donors in the present study?

Thank you for your comment. All kidney transplants were performed from brain death donors. This was specified in the text (line 259)

-       What were exclusion criteria for giving the drug?

The exclusion criteria were a known intolerance, or a side effect related to a previous treatment with iloprost. This was specified in the manuscript (lines 294-5)

-       The drug did not have any affect on the long term (10 year) graft survival, patient survival. Neither it lead to any different acute rejection rates in short term. Please comment on these.

As known, the long term outcomes of kidney transplantation are related to many factors, including the patient comorbities, the chronic rejection and the de novo glomerulonephritis, which could not be influenced by the administration of iloprost

-       Improvement in DGF did not translate into any difference in acute rejection rates. Please comment on possible reason for same.

In our centre, we don’t perform protocol kidney biopsy in kidney transplant recipients, and this practice could have missed some sub-clinical acute rejection episodes in patient with DGF. This was specified in the limits of the study

-       Please give details of sensitization status of each group

This data was inserted in the table

-       Give details of mean HLA mismatches

This data was inserted in the table

-       “Data was analyzed retrospectively but the study was conducted in prospective fashion” is a confusing statement

Thank you for your comment. The sentence was removed.

-       Line no.317 – mentions that acute rejection was based on the basis of graft sonography appearance of worsening of graft function. Please remove this line. There are no conclusive ultrasonographic findings to diagnose acute rejection. It is diagnosed by increasing serum creatinine

Thank you for your comment. I completely agree with the reviewer and the sentence was modified according to your suggestion.

-       What was the mean tacrolimus level between the two groups?

The mean level of tacrolimus was not reported, since the follow-up time is too long.

Reviewer 2 Report

Comments and Suggestions for Authors

The current manuscript describes the injection of iloprost, a prostaglandin analog in the patients who recently received the kidney transplant. The authors find that the continuous injection of iloprost significantly delays the graft function as compared to controls. 

The novelty is good and the  study is properly designed. One small suggestion is the authors might want to elaborate a bit more on the potential side effects of iloprost injection on liver and other organs, since its a high dose of iloprost. 

Also, the authors present data for hundreds of months which is good since these effects become visible after a gap of couple of years. These two points - the concentration of dose of iloprost and the times of injection can be discussed further to complete the manuscript. 

Author Response

Dear Reviewer,

I would like to thank you very much for your valuable comments that will improve significantly the scientific value of the manuscript. Please, you will find below the replies to your comments. I hope to have met all your criticisms satisfactorily.

Best regards

The current manuscript describes the injection of iloprost, a prostaglandin analog in the patients who recently received the kidney transplant. The authors find that the continuous injection of iloprost significantly delays the graft function as compared to controls. 

The novelty is good and the  study is properly designed. One small suggestion is the authors might want to elaborate a bit more on the potential side effects of iloprost injection on liver and other organs, since its a high dose of iloprost. 

Also, the authors present data for hundreds of months which is good since these effects become visible after a gap of couple of years. These two points - the concentration of dose of iloprost and the times of injection can be discussed further to complete the manuscript. 

Thank you for your comments. All transplant recipients were regularly monitored even for liver function, and only a minority of them reported a side effect that caused the discontinuation of the therapy. Moreover, it is not easy to differentiate the long-term liver toxicity from iloprost from that related to immunosuppressive therapy but, generally, iloprost was well tolerated. A brief comment on the side effects observed in our patients was inserted in the manuscript.

Reviewer 3 Report

Comments and Suggestions for Authors

This article performs a randomized, placebo-controlled, single-center study to evaluate the efficacy of prostaglandin-analogue iloprost on preventing DGF in kidney transplantation recipients. This study provided a new potential prevention for DGF in kidney transplantation recipients. However, there are still some concerns that need to be addressed (see below).

Major Concerns:

Results

1.     Line 99: Please use a table to display the results of univariate analysis. If the author feels that this part of the results is unnecessary in the main text, you can display them in the supplementary materials.

2.     There are differences between the normal value of creatinine and the calculation formula for eGFR between men and women, and the proportion of men in the treatment group is significantly higher than that in the control group. Therefore, in addition to comparing creatinine at different time points, eGFR should also be compared in Table 3.

Materials and Methods

1.     Please describe the specific method of randomization in this section

Minor Concerns:

Introduction

1.     In the introduction section, briefly introduce iloprost, including the mechanism through which it may prevent DGF.

Results

1.     Line 75: add standard deviation for the mean BMI.

2.     Line 80: According to the experience of clinical research, the dialysis time of subjects should not conform to the normal distribution. If this is the case, dialysis time for the overall cohort should be expressed as median [quartiles] rather than mean ± standard deviation. The same concern also exists in the "Time on dialysis (months)" row in Table 1

Discussion

1.     Line 178-179: Do not restate the results of other studies in the Discussion section, just state the conclusion.

2.     Line 190-195: These are results, not discussion.

Comments on the Quality of English Language

This article performs a randomized, placebo-controlled, single-center study to evaluate the efficacy of prostaglandin-analogue iloprost on preventing DGF in kidney transplantation recipients. This study provided a new potential prevention for DGF in kidney transplantation recipients. However, there are still some concerns that need to be addressed (see below).

Major Concerns:

Results

1.     Line 99: Please use a table to display the results of univariate analysis. If the author feels that this part of the results is unnecessary in the main text, you can display them in the supplementary materials.

2.     There are differences between the normal value of creatinine and the calculation formula for eGFR between men and women, and the proportion of men in the treatment group is significantly higher than that in the control group. Therefore, in addition to comparing creatinine at different time points, eGFR should also be compared in Table 3.

Materials and Methods

1.     Please describe the specific method of randomization in this section

Minor Concerns:

Introduction

1.     In the introduction section, briefly introduce iloprost, including the mechanism through which it may prevent DGF.

Results

1.     Line 75: add standard deviation for the mean BMI.

2.     Line 80: According to the experience of clinical research, the dialysis time of subjects should not conform to the normal distribution. If this is the case, dialysis time for the overall cohort should be expressed as median [quartiles] rather than mean ± standard deviation. The same concern also exists in the "Time on dialysis (months)" row in Table 1

Discussion

1.     Line 178-179: Do not restate the results of other studies in the Discussion section, just state the conclusion.

2.     Line 190-195: These are results, not discussion.

Author Response

Dear Reviewer,

thank you very much for your valuable comments that will improve the scientific value of the manuscript. Please, you will find below the replies to your comments. I hope to have met all your criticisms satisfactorily.

Best Regards.

Reviewer 3

This article performs a randomized, placebo-controlled, single-center study to evaluate the efficacy of prostaglandin-analogue iloprost on preventing DGF in kidney transplantation recipients. This study provided a new potential prevention for DGF in kidney transplantation recipients. However, there are still some concerns that need to be addressed (see below).

Major Concerns:

Results

  1. Line 99: Please use a table to display the results of univariate analysis. If the author feels that this part of the results is unnecessary in the main text, you can display them in the supplementary materials.

Thank you for your comment. The results of the univariate analysis were reported in Table 1, and this was better specified in the manuscript.

  1. There are differences between the normal value of creatinine and the calculation formula for eGFR between men and women, and the proportion of men in the treatment group is significantly higher than that in the control group. Therefore, in addition to comparing creatinine at different time points, eGFR should also be compared in Table 3.

Thank you for your comment. The eGFR at different time points have been added in table 3.

Materials and Methods

  1. Please describe the specific method of randomization in this section

The method of randomization was inserted in the methods section

Minor Concerns:

Introduction

  1. In the introduction section, briefly introduce iloprost, including the mechanism through which it may prevent DGF.

a brief comment on the rationale of using iloprost for the prevention of IRI was inserted in the introduction.

Results

  1. Line 75: add standard deviation for the mean BMI.

The standard deviation was inserted in the results section

  1. Line 80: According to the experience of clinical research, the dialysis time of subjects should not conform to the normal distribution. If this is the case, dialysis time for the overall cohort should be expressed as median [quartiles] rather than mean ± standard deviation. The same concern also exists in the "Time on dialysis (months)" row in Table 1

Thank you for your comment. In most studies evaluating the impact Of dialysis time on different outcomes in kidney transplant recipients the dialysis vintage was expressed  as mean ± standard deviation.

Discussion

  1. Line 178-179: Do not restate the results of other studies in the Discussion section, just state the conclusion.

 This sentence was modified

  1. Line 190-195: These are results, not discussion.

The sentence was modified

Round 2

Reviewer 3 Report

Comments and Suggestions for Authors

The quality of the revised manuscript has been significantly improved, but there is still a small problem: Table 1 seems to be an comparison of a series of indicators between two groups. This should not be called a univariate analysis. The results of univariate analysis should present the OR value of each single indicator in distinguishing whether a patient develops DGF.

Author Response

The quality of the revised manuscript has been significantly improved, but there is still a small problem: Table 1 seems to be an comparison of a series of indicators between two groups. This should not be called a univariate analysis. The results of univariate analysis should present the OR value of each single indicator in distinguishing whether a patient develops DGF.

Thank you very much for your comment and I agree with the reviewer that the definition is confusing: the table and the manuscript have been revised accordingly